# Te Maramataka—An Indigenous System of Attuning with the Environment, and Its Role in Modern Health and Well-Being

**DOI:** 10.3390/ijerph20032739

**Published:** 2023-02-03

**Authors:** Isaac Warbrick, Rereata Makiha, Deborah Heke, Daniel Hikuroa, Shaun Awatere, Valance Smith

**Affiliations:** 1Taupua Waiora Māori Research Centre, Auckland University of Technology, Auckland 1142, New Zealand; 2Tohunga, Kaumatua (Māori Elder and Environmental Expert), Kaikohe 0405, New Zealand; 3Te Wānanga ō Waipapa—Māori Studies, University of Auckland, Auckland 1010, New Zealand; 4Manaaki Whenua—Landcare Research, Hamilton 3216, New Zealand; 5Te Ara Poutama—Faculty of Māori and Indigenous Development, Auckland University of Technology, Auckland 1010, New Zealand

**Keywords:** Indigenous health, planetary health, Māori, environment and health, decolonising health, maramataka, Indigenous knowledge

## Abstract

The connection between the natural environment and human health is well documented in Indigenous narratives. The maramataka—a Māori system of observing the relationships between signs, rhythms, and cycles in the environment—is underpinned by generations of Indigenous knowledge, observation, and experimentation. The maramataka enabled Māori and our Pacific relatives to attune with the movements of the environment and ensure activities essential for survival and well-being were conducted at the optimal times. A recent revival of the maramataka in various communities in New Zealand is providing uniquely Indigenous ways to ‘reconnect’ people, and their health, with the natural environment. In a world where people have become increasingly disconnected from the natural environment, the maramataka offers an alternative to dominant perspectives of health. It also provides a mechanism to enhance the many facets of health through an understanding of the human–ecosystem relationship in a uniquely Indigenous way. This conceptual paper (i) highlights a uniquely Indigenous way of understanding the environment (the maramataka) and its connection to health, (ii) discusses the connections between the maramataka and scientific research on health and the environment, and (iii) introduces current and potential applications of the maramataka in improving health and well-being.

## 1. Introduction


**
*‘Whatu ngarongaro te tangata, Toitū te whenua’*
**



*As people disappear from sight, the land remains*


This whakataukī (Māori proverb) refers to the enduring quality of the land when compared to the relatively short duration of human generations. It also provides a glimpse into the way Māori (Indigenous people of Aotearoa—New Zealand) view their relationship with the natural environment. Māori, like many Indigenous peoples, believe strongly that the land, forests, wetlands, rivers, estuaries, and oceans play a major role in shaping health and well-being [1]. While this is true at a physical level, whereby healthy ecosystems provide food and clean water, it also relates to strength of cultural identity and spiritual well-being. As a kaumatua (Māori elder) and tohunga (expert) in Indigenous environmental knowledge, Rereata Makiha states—"Health is in the environment” (personal communication, 16 December 2019). In contrast, colonial views, which are still imbedded within current health discourse [2,3] reflect an extractive relationship with the natural environment, where the environment is often seen as a resource to be exploited [4,5]. Although the body of research exploring environment-based (or ‘nature-based’) health interventions is growing [6,7], public health promotion and practice rarely connect the role of the natural environment to achieving good health [8].

Indigenous peoples tend to view humankind as part of a wider ecosystem where living in balance with the natural environment is necessary to sustaining human life [9,10]. This balance is reflected throughout kōrero tuku iho (traditional narratives passed down through generations), like the whakataukī previously mentioned, as well as in contemporary Indigenous health models, such a Te Pae Mahutonga: A Model for Māori Health Promotion [11] and the Atua Matua Māori Health Framework [12]. The environment was closely observed by our tūpuna (ancestors) who, like Indigenous peoples throughout the world, have relied on environmental tohu (signs and signals) to indicate optimal times for performing certain activities. The maramataka for example, is a uniquely Māori system of observing environmental signs, signals, and rhythms, and aligning daily activities to the optimal conditions. Underpinned by the lunar cycle, from new moon to new moon, the maramataka was used to inform the best time to plant and harvest certain crops, collect shellfish, and direct the course of ocean bound vessels [13]. Despite the use of maramataka being limited to a relative few today, largely because of colonisation, interest in and revival of this knowledge and practice is growing.

We posit that Indigenous knowledge systems, such as the maramataka, still hold immense value in contemporary settings—both for Indigenous and non-Indigenous peoples, particularly in strengthening the connection between health and the environment. Thus, the purpose of this conceptual paper is to (i) present the maramataka as a uniquely Indigenous way of understanding the environment and its connection to health, (ii) discuss the connections between the maramataka and science relating to health and the environment, and (iii) explore future research and potential applications of the maramataka in achieving health, particularly among Indigenous people.

## 2. Maramataka—Attuning Health with the Environment

For Indigenous peoples, the mountains, forests, wetlands, rivers, estuaries, and oceans, as well as astronomical features, are often personified and referred to as having familial and genealogical links to humankind. The organisational framework of whakapapa—a Māori term often translated as ‘genealogy’, but which includes a detailed understanding and ordering of the relationships between all things—demonstrates a deep understanding of land, oceanic, and astrological features of the environment as well as their connection to humankind [14]. Based on multiple generations of observing the interactions between environmental phenomena and people, whakapapa provided a way to store and transmit (often in allegorical form) detailed environmental knowledge [15]. For example, Māori often speak of their whakapapa (lineage) to a particular ancestral mountain or river, features of the environment which are intertwined with our cultural identity. Indigenous interpretations of their relationship with environmental phenomena are often described as sharing a physical ‘lineage’—from mountains and rivers to humankind. Although this may seem strange from the view of dominant colonial epistemologies, it should not be too overwhelming to those with a basic understanding of ecology. For example, rain falls on a mountain; the mountain channels the precipitation into streams and rivers; humans drink the water and eat the fish in these rivers; and the water, and nutrients from the fish become part of every cell and DNA strand within the human body. A similar sentiment can be seen in a well-known quote by Carl Sagan: “The nitrogen in our DNA, the calcium in our teeth, the iron in our blood, the carbon in our apple pies were made in the interiors of collapsing stars. We are made of star stuff” [16]. While Indigenous perspectives have often been ignored or interpreted as romanticised myth by colonialists and scientists [17], the contribution of Indigenous knowledge and Indigenous epistemologies to understanding the environment is slowly gaining acknowledgment [18,19,20].

Among Māori in New Zealand, there has been growing momentum to decolonise health and shift the focus of health promotion and practice toward approaches that are underpinned by kaupapa, tikanga, and mātauranga Māori (Māori values, practices, and knowledge, respectively) [21,22]. An important basis of this movement is the development and implementation of culturally relevant initiatives—uniquely Māori, by Māori and for Māori. Within such initiatives, traditional knowledge and culture are key drivers for change, more so than avoidance or treatment of illness, weight loss, or common health promotional messages [23]. For example, an evaluation by Henwood [24] of cultural-based health interventions in New Zealand highlights the strength and potential of grounding health initiatives in traditional knowledge. The maramataka, grounded in ‘traditional knowledge’, provides a culturally relevant lens in which to view health and a culturally relevant system for organising and delivering health services and aligning lifestyle behaviours to the environment.

As previously mentioned, the moon is the baseline time marker of the maramataka with each of the various stages of the lunar cycle assigned a name, and each lunar month is linked with a star or constellation [25]. Based on detailed observations of ecosystems through millennia, each night, or groups of nights (and their corresponding days), provided guidance of events or activities that were vital for the survival of communities [25]. Such knowledge was not limited to the Pacific however, as many Indigenous and ancient peoples observed the moon, stars, and other celestial movements for an indication of the right time to perform a specific activity. Aboriginal Australians of the Arnhem Land had an intimate understanding of the relationship between the moon and changing tides. Their narratives speak of the moon as it rises being filled with water by the high tide, and then the tide falling as a result of the water ‘running out’ of the moon [26]. Likewise, the location of the Pleiades (known as ‘Matariki’ to Māori) was an indicator of first rains to tribes in Southern Africa and a time to prepare for ploughing [27].

Unlike a Gregorian calendar, which identifies the start and finish of a season on the same given date each year, an Indigenous system of time ‘calibrates’ these divisions according to observations of flora and fauna, the cycle of tides, weather patterns, and the appearance and position of stars or star clusters [28]. The changing path of the sun between its two solstice points was also a key indicator for ngā kaupeka o te tau, a Māori concept of the divisions of the year (seasons) [13,29]. Differing significantly from the four seasons that much of the world recognises, Rereata Makiha, a tohunga (expert) of the maramataka, advises that for those in his region (north of the Kaipara Harbour), the year was made up of “7 matiti phases, 3 wero phases, and 3 takurua phases”, while additional phases were added the further south you went to account for the snow, sleet, and ice unique to southern parts of New Zealand (personal communication, 8 March 2022). Subtle nuances of te reo and whakaaro Māori (Māori language and ways of knowing) are often lost when attempting to translate Indigenous terms to English. These three seasonal terms do not correlate directly with Summer, Autumn, Winter, and Spring. Nevertheless, the term ‘Matiti’ is associated with the warmest period of the year, while the ‘wero’ phases represented a transition to cooler weather, and the colder ‘takurua’ phase of the year. Each of these ‘phases’ was marked by the reappearance, or heliacal rising of certain stars, paired with the observation of specific changes in the environment. For example, Wero-i-te-kokotā (the second wero ‘phase’) is marked by the reappearance of the star by the same name (a star within the constellation Canis Major) and the arrival of the first frost. Living by this system naturally requires a close relationship with, careful observation of, and regular engagement in the natural environment.

Indigenous epistemologies tend to see people as part of a wider ecological and spiritual network [30] so that environmental science, spirituality, and health existed as integrated and inseparable [15]. Atua—A Māori term often translated as ‘god’ or ‘deity’, but which encompass personified forms of the environment—are closely connected to the maramataka. Roberts et al. [31] (2006, p. 1) suggest that the maramataka

“…demonstrate(s) this interweaving of essential economic activities aimed at food procurement, with social activities including ritual observances pertaining to that resource and their guardian atua”.

Ultimately, the maramataka provides a culturally relevant approach to health that is embedded with spiritual aspects of well-being, which shifts the focus of health promotion from biomedical components of sickness to cultural principles valued by Māori and Indigenous peoples globally [31]. What is more, engagement with knowledge systems such as the maramataka provides opportunities for cultural reconnection and the strengthening of identity, both of which have a direct impact on health [32,33].

## 3. Maramataka—Connections between Indigenous Knowledge and Health Science

Global concerns about the impacts of climate change, and the emergence of the COVID-19 pandemic, have provided a timely reminder of the connection between environment and human health. Various levels of the connection between health and the natural environment are well established in academic literature as well, with several studies revealing that the time spent in outdoor spaces is associated with improved psychological [34] and physical well-being [35]. For example, studies on ‘forest therapy’ have shown that increased time in the forest has a positive effect on cognitive and immune function, hypertension, cardiovascular disease, cancer, pain, and blood glucose [36]. Others showed that walking for 90 min through natural environments led to lower levels of negative thoughts about oneself and reduced neural activity associated with mental illness when compared to those walking in urban environments [37].

The literature also highlights the close connection between human health and the rhythms and cycles within the environment. Celestial and environmental phenomena not only influence our ability to provide food but also have a direct effect upon animal/human physiology and behaviour. For example, Sjoden et al. [38] showed that leptin, a hormone associated with satiety (feeling full), was higher in children on days near the full moon compared with days around the half moon, while ghrelin, a hormone associated with feelings of hunger, were lower. Furthermore, Walton et al. [39] (p. 303) state that

“Central to the evolution of photoperiodism in animals is the adaptive distribution of energetically challenging activities across the year to optimize reproductive fitness while balancing the energetic trade-offs necessary for seasonally appropriate survival strategies. The ability to accurately predict future events requires endogenous mechanisms to permit physiological anticipation of annual conditions”.

Inherent in the maramataka is the ‘distribution of energetically challenging activities’ to ensure that energy is used to perform certain activities at the optimal time [13]. What is more, this approach ensures that the environment is given periods of time to recover and replenish from harvesting pressures. For example, if the optimal days for fishing were already known, then energy was not wasted on days which were likely to yield low returns, and these may coincide with days set aside specifically for ‘giving back’ to (replenishing) the environment where no food was taken [40] (p. 73).

In developed countries where lifestyles often revolve around a 9 A.M. to 5 P.M., 5 day-a-week (or more) work routine, human physiological rhythms have become more disordered and more out of sync with the rhythms of their natural surroundings [41]. Although rest and recovery are vital for mental and physical well-being, modern lifestyles rarely include a ‘natural’ variation in high-intensity and lower-intensity activities and sufficient rest. Certain ‘inflexible’ employment situations are associated with burnout and mental illness [42], and West and Bechtold [41] (p. 778) suggest that “erosion of our circadian rhythms is associated with a collection of metabolic problems… disorders involving immune dysfunction, neurobehavioral abnormalities, and cancer”. Accordingly, some companies have begun to experiment with varying length of workdays and daily interactions with nature to improve/maintain employee morale and well-being [43,44]. The maramataka, which encourages the cycling of activities and levels of ‘work’ intensity, could have applications in managing stress and lifestyle by considering the natural ebbs and flows of gene expression, hormone secretion, cellular receptor activity, and other physiological rhythms.

In addition to modern work habits, dietary habits have also changed considerably. With increased globalisation, access to fruits, vegetables, and animals for food no longer ebbs and flows with seasons nor is it restricted to a local climate. Instead, tropical fruits are found in stores in sub-arctic regions, and meat whose availability was determined on migratory patterns is readily available as fresh or frozen in the grocery store. Ultimately, the maramataka is focused on food procurement [31], and one of its main roles was “as a predictive tool for scheduling activities critical to the continued success of hapū and iwi [sub-tribes and tribes] such as fishing and gathering kai moana [sea food]” [45] (p. 7). A similar sentiment is also reflected in the Hawaiian concept of ‘Kilo’—a term associated with making “place-based observations, that enabled our Indigenous ancestors to understand their surroundings well enough to know when and where to gather food in order to sustain themselves for generations” [46] (p. 9). On some days or at certain times of the year, specific sea foods would have been a staple. Other times, diets might have prioritised plant-based eating, and there would also be specific times where fasting was encouraged and aligned with environmental indicators. Those who observe the maramataka are constantly reminded of the connection to the environment through food, while modern day food systems tend to distance people from an understanding of that connection [47]. What is more, those following the maramataka were essentially following a seasonal dietary model that might be considered more ‘balanced’ among many nutritionists today.

As Indigenous scientists, we observe numerous implications, applications, and areas of alignment between the maramataka and various scientific fields that relate to health beyond the few previously mentioned. For example, the field of epigenetics and the study of developmental origins also highlights the role of environment in shaping a person’s gene expression and their response to environmental stimuli such as inadequate or excessive food supply, toxins, or varying psychological stressors [48,49]. What is more, those who advocate for ‘personalised medicine’, which “seeks to identify groups or strata of patients with specific molecular characteristics or other determining factors [to] predict prognosis and response to therapy” [50] (p. 14), may be interested in the maramataka as a ‘personalised’ approach to health—not necessarily personalised to an individual person, but to the environment and ecosystem in which the person is a part. This shift away from person-centred health towards environment or ecosystem-centred health is another defining feature of Indigenous perspectives of health, nutrition, and environmental management [51,52,53]. Maramataka were attuned to meet the needs of differing environments, as can be seen by the dozens of regional, local and even family-specific maramataka known throughout Aotearoa [31].

Although many helpful records and maramataka experts have kept this tradition and practice alive in New Zealand and throughout the Pacific, many aspects of traditional knowledge and their applications have been lost through the ongoing processes of colonisation. Many Indigenous peoples are disconnected from ecosystems which sustained them for generations due to urban drift and confiscation of lands. Indigenous rights to practice and maintain knowledge relating to tātai arorangi (traditional astronomical knowledge) have also been intentionally and systematically dismantled through legislative tools [54,55]. While Indigenous knowledge is often seen as unscientific, Hikuroa [45] (p. 6) affirms that the ‘maramataka is knowledge generated using the scientific method, [but] explained according to a Māori world view’.

Although some continue to see Indigenous knowledge as inferior to ‘science’ [56], the Indigenous wisdom underpinning the maramataka was and continues to be based on many of the aspects valued by ancient and modern scientists. Observation of environmental phenomena recorded by generations of Pacific ancestors in the maramataka, also known as ‘Te Arapo’ in the Cook Islands or ‘Kaulana Mahina’ in Hawai’i, represents a potentially more robust and exponentially longer ‘longitudinal study’ of environmental, agricultural, ecological, and aquacultural science than any conducted by modern research institutions or researchers. The stakes of this ‘longitudinal’ observation were extremely high—in the case of early Polynesian observers, the result of incorrect application would not only result in a diminished reputation among peers (a significant motivator within academic communities) but had implications for the longevity and survival of communities [13].

Nevertheless, the purpose of this article is not to legitimise or validate Indigenous knowledge against Eurocentric scientific standards. In fact, Indigenous knowledge needs no validation, particularly against a colonial standard that has been a cause of significant harm for Indigenous peoples [29,57,58]. Rather, we have highlighted areas of overlap between dual epistemologies, Indigenous knowledge and Western ideologies, with respect to health science and practice. The revival of Indigenous practices, such as the maramataka, is a step toward Indigenous autonomy and self-determination in health [11]. What is more, one aim of Indigenous health research is to decolonise the systems that have oppressed and harmed Indigenous people while creating services and structures that support Indigenous people to determine and lead their own provision and promotion of health [11,59,60]. As Durie [11] (p. 7) suggests,

“No matter how dedicated and expertly delivered, health promotional programmes will make little headway if they operate in a legislative and policy environment which is the antithesis of health, or if programmes are imposed with little sense of community ownership or control”.

We suggest that revitalising and reaffirming the practices associated with the maramataka do all the above.

## 4. Maramataka—A Model for a Healthy Lifestyle

Throughout New Zealand, there are more than 30 known maramataka from various tribal regions [31]. While these different maramataka have much in common, there are variations either in the names of certain days, the order of those days in the ‘calendar’ (Figure 1), or the activities associated with those days. The variation in the details of the maramataka between tribes is no surprise considering the varied landscapes, climates, and environments between those tribes in the sub-tropical north and those in the sub-arctic south; those who occupied coastal regions and those who dwelt inland. Unlike commercialised health programmes and modern health initiatives, the maramataka was never designed to be prescriptive in nature. The details and activities associated with each day were tailored to place, while being dynamic and adaptable according to observations of varied and changing environments. The system was constantly tested and validated according to observations of the changing environment [45].

Although there exist several systemic and structural challenges to restoring this practice, a handful of groups and communities are revitalising the maramataka and other Indigenous environmental health practices in their own regions [61,62]. As is common for Indigenous initiatives, however, these are rarely outlined in scientific literature so that ‘peer-reviewed’ research and ‘evidence-based practice’ about the maramataka is very limited. On the other hand, Indigenous peoples may be resistant to engage with scientific research, institutions, and literature, particularly when traditional knowledge is interpreted and critiqued by non-Indigenous scholars, and those who treat such knowledge as nothing more than relics of interest from a bygone past [63,64]. Worse still, Indigenous knowledge and histories more broadly were deliberately diluted and purposely adapted to align with the political agendas of colonisation, or as Moana Jackson describes “the construction of a particular reality” and “the imposition of a philosophical and myth-making construct” [65] (p. 92). Nevertheless, the tides seem to be changing, with an increasing number of Indigenous academics and non-Indigenous allies highlighting the value of traditional knowledge, approaches, and practice for modern health settings [10,66,67,68].

Here, we highlight some Indigenous-driven initiatives (mostly from non-peer-reviewed sources), such as one Māori scholar conducting doctoral research on the association between phases of the maramataka and suicide [61], while another’s research is identifying the role of the maramataka in woman’s health and childbirth [69]. Another initiative driven by communities in the Northland region looks to the maramataka and traditional narratives to guide the monitoring of certain fish species and the replenishment of other food sources in the region [70]. Others in this same region have utilised the maramataka in managing organisations, from hospitals to government, where strategising and planning are conducted on ‘high energy’ days and assessments are completed on days associated with ‘lower energies’ [71]. Others have structured exercise training programmes around the maramataka, varying activities according to different stages of the lunar cycle [72]. For example, high-intensity workouts scheduled for days of the maramataka associated with ‘higher energies’ such as Rakaunui—the period during a full moon—while during Whiro, the period during the new moon which can be associated with ‘lower energies’, low-impact activities such as stretching are performed [72].

Perhaps the greatest concern of the authors, in presenting ideas for how the maramataka could be used to manage lifestyle, is the risk of the maramataka being adopted or co-opted in an overly prescriptive and universalistic national ‘health programme’ or potentially exploited as an Indigenous health product. While a common practice in Western health is to prescribe a programme—of exercise, diet, smoking cessation, or a multitude of other behaviours—aligning health with the maramataka and therefore culture, observation, and the environment is a shift away from such prescriptive health practices. While health professionals obviously have a place, an underlying component of the maramataka is self-determination, reclaiming autonomy and power over one’s own health so that health advice fits within one’s own framework of observation [11,67]. Ultimately, the maramataka is an enduring legacy that ensures our own well-being is maintained and importantly connects us to our collective responsibility of ensuring future generations flourish [73].

## 5. Maramataka—Past, Present, and Future Applications

The sources already discussed, both modern and ancient, suggest it unwise to continue a singular pursuit of improving human health without at the same time pursuing a connection to, and observation of, our natural environment [74]. While it has become common to hear mention of the ‘human footprint on the environment’, perhaps we need to start talking more about the environment’s footprint upon us.

Although there is a significant body of literature about health and the environment from the perspectives of non-Indigenous researchers [34,35,36,37,38], much less is known about how spending time in and engaging with natural environments impacts the health of contemporary Indigenous peoples. Research on the maramataka could help us better articulate a planetary health approach in a way that reflects Indigenous perspectives of health—some of these benefits could be unique to Indigenous peoples, considering the spiritual and physical connections to specific local ecosystems, and their links to cultural identity [32].

While Māori communities and groups around New Zealand are leading the way in revitalising these practices, support of maramataka initiatives at policy, funding, research, and government levels could contribute to achieving Indigenous health aspirations at a community level. The maramataka has relevance beyond health and may prove useful in addressing wider determinants of health such as education, environmental restoration, economic development, etc. The aim of research projects and programmes would not be to validate Indigenous knowledge but rather to understand the benefits of the maramataka and how it can best be applied and practiced in a modern-day context. As mentioned earlier, the maramataka, and indeed Māori knowledge in general, is adaptable and provides a framework for which applications may change in response to need [10,45,75]. Future research, including community-driven inquiry, should continue to identify and explore novel applications of the maramataka. While such a research agenda should be led by Māori or relevant Indigenous communities, it may benefit integrating modern scientific approaches as well. Such an interface between Western approaches to science and Indigenous knowledge provides a holistic perspective that ensures that multiple aspects of health and well-being are considered by health practitioners [76,77].

As we have interacted with other peoples and cultures throughout history, Māori have regularly adopted and adapted new and ‘foreign’ technologies while still maintaining our cultural identity and ways of being [78]. Future research could also help us identify new ways that the maramataka can work alongside non-Indigenous forms of knowledge creation, science, and technology to improve the health of people and the environment. Doing so, without losing the connection to Indigenous perspectives, values, and worldviews (those which underpin the maramataka), is essential. Modern technologies and scientific tools may prove useful alongside Indigenous systems but only if Indigenous peoples are also empowered to implement those tools alongside their own practices and epistemologies.

Regardless of the research projects that emerge around the maramataka, it is essential to keep in mind the ethos of conducting research by Indigenous peoples, for Indigenous peoples (by Māori, for Māori) [60]. This does not exclude non-Indigenous researchers, health, or environment experts from participating or contributing. On the contrary, non-Indigenous allies and teammates provide valuable expertise and support in decolonising health at an interface between Indigenous knowledge and science [76]. Rather, it means that Indigenous aspirations, and more specifically the aspirations of local iwi and hapū (tribes and extended family groups) for healthy people and a healthy environment, remain the primary goal [60].

## 6. Conclusions

Helm et al. [79] (p. 1) suggest that “consideration of Homo sapiens as principally a ‘seasonal animal’ can inspire new perspectives for understanding medical and psychological problems”. Likewise, the use of the maramataka in modern days reminds us of a real connection with the environmental rhythms that peoples around the world have observed and been attuned with for countless generations. Not underpinned by a Western, neo-liberal lens, or influenced by colonial, commercial, or racist perspectives, the maramataka provides a culturally relevant guide to organise one’s lifestyle and align with balance and good health. We argue, that for Māori at least, a people who have not been well served by dominant Euro-Western health approaches, aspirations of good ‘health’ and well-being will not be achieved unless there is a (re)connection to the natural cycles and rhythms of our environment. Underpinned by Indigenous knowledge and evidence observed and gathered over thousands of years, and in various locations from the tropical Pacific islands to the sub-tropical and alpine terrains of Aotearoa–New Zealand, the maramataka provides a uniquely Indigenous lens to approach healthy living and lifestyle change through strengthening connections to, and place in, the environment.

## Figures and Tables

**Figure 1 ijerph-20-02739-f001:**
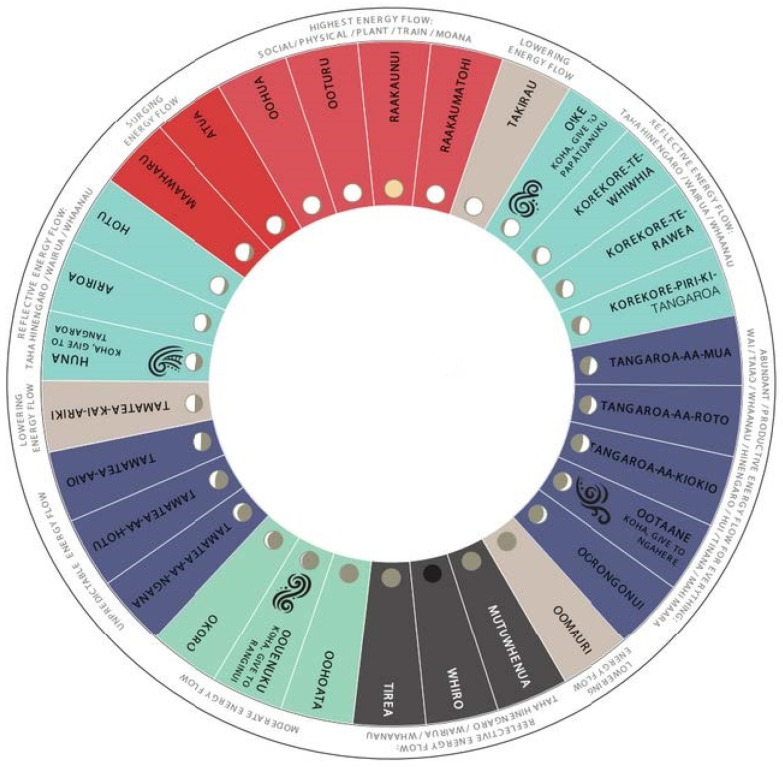
A contemporary visual representation of one maramataka from the Manukau region. In this example, 30 days are outlined with their respective names and correlating lunar phase, while the outer ring outlines the differing levels of activity or ‘energy’ relating to that period. Designed and developed by Ayla Hoeta.

## Data Availability

No new data were created or analyzed in this study. Data sharing is not applicable to this article.

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
