# Peer review of "Te Maramataka—An Indigenous System of Attuning with the Environment, and Its Role in Modern Health and Well-Being"

_ijerph, 2023, doi:10.3390/ijerph20032739_

Round 1

Reviewer 1 Report

Affirming the knowledge Māori have once had and are now working to reclaim re: māramataka, is key to our collective evolution. This article lifts and honors these practices and principles into mainstream awareness so that an older definition of healing can be maintained. Awareness of natural systems is by itself an epistemology. 

Author Response

We appreciate the reviewer's positive review and encouraging feedback and agree that awareness of natural systems (as reflected in the maramataka) is an epistemology in itself – we hope this manuscript contributes to the work of others in highlighting this important point.

Reviewer 2 Report

Having read the paper, I consider it relevant because the topic is hinged around traditional means of keeping people in tune with nature. However, the paper lacks scientific depth and is poorly structure (but well argued).

The article presents an exploratory (or expository) narrative that, despite understandable and real, is not build scientifically on a plausible path. This would a great newspaper/magazine article but not journal article - except it is woven within known theory, literature and reproducible methodology. I can also go for an information(communication paper in an academic journal.

Author Response

We appreciate the reviewer's time in reviewing this manuscript and their encouragement about the relevance of the topic. We have found it difficult as author's to address the main suggestions made here, without more specific details or examples relating to 'scientific depth and structure'.

This manuscript, as a conceptual article, was structured to identify ways of knowing that are often missing from scientific understanding and practice of environmental/planetary health (namely an Indigenous epistemology), and to provide a uniquely Indigenous example, the maramataka, as a way to advance the environmental/planetary health field. Many references are intentionally made to Indigenous forms of knowledge dissemination (narratives, proverbs etc.), along side academic literature, as a way of challenging common perspectives and proposing alternative approaches to living a healthy lifestyle in line with the natural environment.

Reviewer 3 Report

The abstract is very well written, leading into the manuscript body, which is well-written overall.

The manuscript is successful in laying the conceptual foundations for the maramataka with respect to Māori and indigenous health. The designated sections make sense conceptually, and lead to a key point that further research on the application of the maramataka is needed. The question of the extent to which people are/will be enabled, in the context of our colonial, neoliberal environment, to practise/live/work according to the rhythms of the maramataka, remains.

Page 3 lines 129 – 130 – define/translate Matiti, wero and takurua?

Page 4 line 185 West and Bechtold

Page 5 lines 222-4 – the point about a personalised approach is excellent. “Not necessarily personalised to an individual person, but to the environment and ecosystem in which the person is a part” – this gets to individualism from an Indigenous (relational) perspective.

Final point/sentence in second to last paragraph (lines 229 – 231) – I’m not sure this is needed, or perhaps it should be placed elsewhere in the paragraph?

Author Response

We appreciate the Reviewer's encouraging feedback and helpful suggestions to enhance the polish of the finished manuscript.

  • Although it can be difficult to define/translate Māori terms directly into English (particularly when talking about ‘seasons’ with no English equivalent), we have added a footnote providing an explanation and interpretation of these terms. We have also provided an example, in the text, of the indicators which mark these phases to help the reader better understand this point.
  • The West and Bechtold citation has been amended
  • We appreciate the encouraging feedback about an (Indigenous) personalised approach.
  • The sentence – ‘Nevertheless, it is beyond the scope of this article to delve into an exhaustive list of ways that the maramataka connects with various scientific fields.’ – has been deleted.